# The Impact of Swine Manure Biochar on the Physical Properties and Microbial Activity of Loamy Soils

**DOI:** 10.3390/plants11131729

**Published:** 2022-06-29

**Authors:** Muhammad Ayaz, Dalia Feizienė, Virginijus Feiza, Vita Tilvikienė, Edita Baltrėnaitė-Gedienė, Attaullah Khan

**Affiliations:** 1Lithuanian Research Center for Agriculture and Forestry, Institute of Agriculture, Instituto al. 1, LT-58344 Kėdainiai, Lithuania; dalia.feiziene@lammc.lt (D.F.); virginijus.feize@lammc.lt (V.F.); vita.tilvikiene@lammc.lt (V.T.); 2Institute of Environmental Protection, Vilnius Gediminas Technical University, LT-10223 Vilnius, Lithuania; edita.baltrenaite-gediene@vilniustech.lt; 3School of Forestry, Northeast Forestry University, Harbin 150040, China; khan52@aup.edu.pk

**Keywords:** biochar, carbon source utilization, soil indices, soil hydraulic conductivity, soil porosity

## Abstract

Biochar has been proven to influence soil hydro-physical properties, as well as the abundance and diversity of microbial communities. However, the relationship between the hydro-physical properties of soils and the diversity of microbial communities is not well studied in the context of biochar application. The soil analyzed in this study was collected from an ongoing field experiment (2019–2024) with six treatments and three replications each of biochar (B1 = 25 t·ha^−1^ and B0 = no biochar) and nitrogen fertilizer (N1 = 160, N2 = 120 kg·ha^−1^, and N0 = no fertilizer). The results show that biochar treatments (B1N0, B1N1, and B1N2) significantly improved the soil bulk density and total soil porosity at different depths. The B1N1 treatment substantially enhanced the volumetric water content (VMC) by 5–7% at −4 to −100 hPa suction at 5–10 cm depth. All three biochar treatments strengthened macropores by 33%, 37%, and 41%, respectively, at 5–10 cm depth and by 40%, 45%, and 54%, respectively, at 15–20 cm depth. However, biochar application significantly lowered hydraulic conductivity (HC) and enhanced carbon source utilization and soil indices at different hours. Additionally, a positive correlation was recorded among carbon sources, indices, and soil hydro-physical properties under biochar applications. We can summarize that biochar has the potential to improve soil hydro-physical properties and soil carbon source utilization; these changes tend to elevate fertility and the sustainability of Cambisol.

## 1. Introduction

In recent years, biochar has been used extensively as a soil conditioner to improve soil quality [1]. Soil physical conditions directly influence soil fertility by determining water retention capacity, aeration, and soil permeability, which tend to improve soil productivity [2]. Several studies indicated that good soil structure, porosity, hydraulic conductivity, and specific gravity created favorable conditions for essential soil microbial growth and enabled the more efficient use of water and nutrients in the soil profile [3,4].

Moreover, the addition of biochar improved nutrient and water retention and increased root growth substantially compared to degraded soils with poor physical properties [5]. Biochar amendment improved soil hydraulic conductivity by decreasing soil bulk density and increasing soil porosity [6]. However, some studies found that biochar has little effect or even negative effects on soil physical properties. Previously, there was no proof to support the claim that biochar amendment influenced soil porosity by direct pore contribution, the creation of accommodation pores, or ameliorations in aggregate stability [7]. The application of biochar lowered pore connectivity and the number of macropores in a wheat–rice rotation system [8]. Such variations signify that the effects of biochar on soil structure and hydraulic properties are unclear.

Numerous reports have stated that biochar application substantially enhances microbial activity in soil [9,10]. In theory, the application of almost antiseptic biochar would dilute the diversity of soil microorganisms [11]. Macropores and the large surface area of biochar tend to contribute to the loss of volatiles during pyrolysis, which creates favorable conditions for soil microbiota in the long term [12]. Previous reports have also indicated a decline in microbial abundance and mycorrhizal diversity with the addition of biochar; these studies specified conditions that depended upon the nitrogen (N) and phosphorus (P) levels in soil [13], due to a lower adverse effect caused by the extreme content of mineral elements in biochar [14,15]. Certainly, microbial abundance is very sensitive to ecological factors such as the physicochemical characteristics of soil [16]. Gul (2016) reported that biochar particles could intervene as an unconventional niche for soil microbes in relation to soil water content, pH, aeration, and other physicochemical properties [17]. Furthermore, it was added that biochar-absorbed organic carbon from the contiguous soil might be directly consumed as an energy source by soil microorganisms [18]. In addressing all these elements, it is necessary to measure the effect of biochar on soil microbial activities with respect to biochar type and soil properties. However, data regarding the mechanisms of how biochar affects microbial diversity and abundance are still lacking. Soil microbes are considered as executors in the soil environment [19], and their distribution, abundance, and diversity are vital to ecosystem resources and soil function [20]. The plate-counting procedure can be employed to count only a low percentage of the entire soil microbial abundance; however, it has also been frequently used and reflects microbial biomass reliably. This is a legitimate procedure for counting the bacteria and organisms with specific functions that can live in artificial media. The diversity of soil microorganisms is another point of interest in the soil ecology context and is considered an indicator of soil health [21].

There are several methods for studying the phylogenetic diversity and abundance of soil microbial communities, which relate to the phospholipid fatty acids of the microbial membranes [22]. Additionally, the correlation between the fingerprint [23] and the environmental factors of the denaturing gradient gel electrophoresis profile can be established by redundancy analysis. Generally, microbial syndicates with high genetic variability have the capacity to consume more diverse carbon (C) sources; this can be evaluated by their community-level physiological profile, which leads to their C metabolic potential. The complex correlation of different soil functions makes analyses difficult [24]. The relationships between the chemical and physical properties of soils amended with biochar and their effects on soil biota are poorly understood. Soil microbial functions and the soil pore structure influence some soil physical properties and determine the retention, transport, and supply of soil moisture and, therefore, crop yield through their interactions [25].

Thus, we hypothesized that a swine-digestate-derived biochar amendment could influence soil microbial diversity and abundance and also the utilization of soil carbon source (SCS) (28 substrates) by influencing the hydro-physical properties of soil. Furthermore, the microbial potential of SCSs may vary in the presence of biochar carbon sources. To test the above hypotheses, we examined the hydro-physical soil properties, microbial abundance, and SCS utilization of biochar-amended loamy Cambisol (sand (2.0–0.05 mm), 50.1%; silt (0.05–0.002 mm), 31.1%; and clay (<0.002 mm), 18.8%) under moderate climatic conditions.

## 2. Materials and Methods

### 2.1. Biochar and Soil

Swine manure digestate was collected from an active animal farm. The manure digestate was air-dried for 48 h and manually ground. The feedstocks were pyrolyzed at 550 °C in a cylindrical furnace for 5–6 h under anaerobic conditions to produce biochar [26]. The performance of the feedstock during thermal decomposition was tested through thermo-gravity analysis (TGA) with a thermal analyzer, namely the Netzsch Jupiter STA 449 F3, at the Lithuanian Energy Institute [27]. During the TGA process, the pyrolysis process was applied, with a 35 °C/min heating rate in the temperature range 40–900 °C with 9.6 ± 0.32 of the feedstock sample. To create an inert atmosphere, N_2_ carrier gas (60 mL/min) was used. *Endocalcari-epihypogleyic* Cambisol soil (WRB, 2014) was used in this study; it was obtained from 0 to 20 cm depth in a farmland field of the Institute of Agriculture (55°23′49″ N and 23°51′40″ E) at the Lithuanian Research Centre for Agriculture and Forestry. The soil samples were air-dried, homogenized, and meshed through a 2 mm sieve before use, and the physicochemical properties of the soil and biochar are given in Table 1.

### 2.2. Experimental Design

A three-factorial randomized complete block design (RCBD) field experiment was designed with six treatments and three replications each. The combination of the treatments was as follows: B0N0 (no biochar + no Nitrogen fertilizer), B0N1 (no biochar + 160 kg·ha^−1^ Nitrogen fertilizer), B0N2 (no biochar + 120 kg·ha^−1^ Nitrogen fertilizer), B1N1 (25 t·ha 25 t·ha^−1^ biochar + 160 kg·ha^−1^ Nitrogen fertilizer), B1N2 (25 t·ha 25 t·ha^−1^ biochar + 120 kg·ha^−1^ Nitrogen fertilizer), and B1N0 (25 t·ha^−1^ biochar + no Nitrogen fertilizer). Ammonium nitrate was used as an N fertilizer. The biochar was broadcast and shallowly incorporated into the soil surface during pre-sowing tillage. The main crop was spring barley (*Hordeum vulgare* L.), which was sown in April 2020 and harvested in August 2020. The weather conditions at the experimental site are given in Figure 1 and were obtained from a meteorological station located 0.5 km away from the experimental site.

### 2.3. Chemical Analysis

The physicochemical properties of the biochar and soil were analyzed by standard laboratory methods. Soil and biochar (pH) and EC analyses were performed using a 1:5 (vol vol^−1^) soil mixture in a 1 M KCl solution [28] and an extract to distilled water [29], respectively. Cation exchange capacity was determined with an updated ammonium-acetate method [30]. Inductively coupled plasma atomic emission spectrometry (Perkin Elmer ICP-OES, Waltham, MA, USA) was used for measuring the DTPA extractable nutrients P, K, Ca, and Mg [31]. The contents of total nitrogen (TN), available phosphorus (PA-L), and potassium (KA-L) were measured using a reference method [32]. The data for biochar ash content, moisture, volatiles, and residual mass were obtained from TGA.

### 2.4. Hydro-Physical Soil Analysis

For the analysis of soil water retention characteristics and pore-size distribution, undisturbed soil samples were taken in stainless steel cylinders (51 mm high and 53 mm in diameter) from each treatment. Water retention properties were studied at −4, −10, −30, and −100 hPa (in a sand box) and at −300 hPa (in a sand–kaolin box). Water content was determined at −15,500 hPa of suction using sieved soil samples [33]. The water content levels at −100 hPa and at −15,500 hPa were considered the field capacity (FC) and the permanent wilting point (WP), respectively. The amount of water between these two suctions was regarded as plant-available water (PAW) content. Soil cores were stored in the refrigerator at a constant temperature (2 °C). The same samples were used for the analysis of soil bulk density (BD), total air-filled porosity, and saturated hydraulic conductivity. HC was determined with a laboratory permeameter (Eijkelkamp, The Netherlands) by the constant head method [34].

### 2.5. Microorganism Community-Level Substrate Utilization Pattern Analysis Using Biolog^®^ Ecoplate

The Biolog system and the Biolog^®^ EcoPlate procedure were used to determine 31 types of carbon and the metabolic functional diversity of soil microorganisms; this system was particularly meant for community analysis and microbial environmental research [35]. For this purpose, fresh soil samples were collected from each plot. Samples were air-dried, ground, and meshed through a 2 mm sieve. A 10 g dried soil sample was collected from each treatment and mixed with 90 mL of distilled water in a 250 mL flask at 250 rpm for 30 min in the rotary shaker. From each 10–3 diluted suspension, 150 μL was added into a 96-well Biolog^®^ EcoPlate (Biolog, Hayward, CA, USA); these samples comprise three replications of 31 widely useful carbon sources, and one was regarded as a control treatment (without a C source). Data for the absorbance-incubated plates were recorded at 590 nm (dual-wavelength data: OD590–OD750) every 24 h at 25 °C for periods of 24, 48, 72, and 96 h [36]. The resulting data of each well, i.e., the color changes from the carbon utilization of the soil microbes, were investigated in Microlog 4.01. For the estimation of the integral fingerprinting of carbon source utilization, the average well color development (AWCD) was used for each microplate well per reading time [37]. The EcoPlate readings at 15 d were used to analyze the Shannon index (H), richness (S), Simpson index (D), and McIntosh index (U), which evaluated the diversity, richness, number, and evenness of the soil microbes, respectively [38].

### 2.6. Calculation of the Species Diversity Indices

Species richness is regarded as the number of species in a sample, whereas the distribution of individuals among the recorded species is considered species evenness. The information needed to describe every species of the community is known as Shannon’s index; it is calculated using the following Equation (1):(1)$$H=\overset{s}{\underset{i}{\sum }}=1\;P_{i}\;\log_{pi}$$
where *s* is the total number of species and $P_{i}$ is the proportion of all individuals in the sample that belongs to species *i*.

Simpson’s index (D) assesses the contingency that two species randomly chosen from a sample belong to the same species. See Equation (2):(2)$${\mathrm{Simpson}}^{’}s\;\mathrm{index}\;\left(D\right)=\sum {\left(n/N\right)}^{2}$$
where *n* represents the total number of organisms of a specific species and *N* is the total number of organisms of all species.

Species richness (R) is regarded as the number of species in a particular area and is calculated using the following Equation (3):(3)(R)=S−1/Iog(N)
where *N* = the total number of individuals in the sample and S = the number of species recorded, and U is represented by Expression 4, also known as the Mclntosh index:(4)$$U=\sqrt{\sum n_{i}^{2}}$$
where *n* (*i*) represents the number of individuals in the *i*th species, the sum is that of all the species, and U is the Euclidean distance of the community from the origin [39].

Species evenness, described by Magurran (1988) [40] as another index of diversity, is calculated using the diversity index, as in Equation (5):(5)$$\mathrm{Species}\;\mathrm{Evenness}=H/H_{max}$$
where H’ = Shannon’s index and *H_max_* = lnS, where S is the number of species present in the community.

### 2.7. Statistical Analysis

Three-way analysis of variance (ANOVA) was used to compare the effects of factors A (25 t·ha 25 t·ha^−1^ biochar and without biochar), B (160 kg·ha^−1^, 120 kg·ha^−1^, and without N fertilizer), and C (depths 5–10 and 15–20 cm; timing 24, 48, 72, and 90 h) and their interactions on the soil hydro-physical properties, carbon sources, and indices. The homogeneity of variances was tested with Levene’s test. Normality was assessed with the Shapiro―Wilk and Durbin―Watson tests. A post-hoc Tukey HSD test was used to analyze the differences between treatments. Pearson’s correlation coefficients were calculated to explore the interrelations between and within the hydro-physical properties and soil microbial abundance and also between B0 and B1 under N0, N1, and N2 conditions. The Pearson’s correlation analysis was performed using the corrplot package in R [41]. Redundancy analysis (RDA) was used to compare the interrelations between the carbon sources and soil physical properties. RDA analysis was performed with the vegan package in R [42]. The Pearson’s correlation analysis (PCA) was performed with the PCA package in R [43]. All statistical analyses were performed using SPSS v. 25.0 (IBM Inc., Armonk, NY, USA). Sigma Plot v. 12.2 (Systat Software Inc., San Jose, CA, USA) was used for graphical representation.

## 3. Results

### 3.1. Soil Bulk Density and Total Porosity

Dry soil bulk density and total soil porosity in the investigated treatments during the month of May and August are summarized in Figure 2. The application of 25 t·ha^−1^ of biochar alone (B1N0) and biochar with 120 kg·ha^−1^ of N fertilizer (B1N2) significantly (*p* = ≤0.05) enhanced soil BD by 10–12% at 5–10 cm soil depth in the month of May (Figure 2A); in August, biochar alone and biochar with 160 kg·ha^−1^ of N fertilizer (B1N1) significantly (≤0.05) enhanced soil BD by 8–10% compared to non-biochar treatments (Figure 1B). Similarly, treatments B1N0 and B1N1 substantially (*p* = ≤0.05) increased soil BD at 15–20 cm depth by 9–10% in May compared to non-biochar treatments, whereas, in August, treatment B1N0 at 15–20 cm soil depth significantly (*p* = ≤0.05) increased soil BD compared to other treatments (Figure 2A). Soil porosity was recorded as being substantially (*p* = ≤0.05) higher in non-biochar treatments at 5–10 cm depth in May (Figure 2C); however, in August, 25 t·ha 25 t·ha^−1^ of biochar with 120 kg·ha^−1^ of N fertilizer significantly (*p* = ≤0.05) increased soil porosity at both 5–10 cm and 15–20 cm depths by 17% and 15%, respectively, compared to non-biochar treatments (Figure 2D). Factors A, B, and A*B showed significant (*p* = ≤0.05) impacts on soil BD and TP, whereas factor A*B*C recorded non-substantial results. Thus, the overall effect of biochar with and without N fertilizer was positive with respect to BD and TP.

### 3.2. Volumetric Soil Water Content

In our current study, the volumetric soil water content (VWC) at −4 to −100 hPa suction at 5–10 cm depth was recorded as being higher (5–7%) under the application of 25 t·ha 25 t·ha^−1^ of biochar and 160 kg·ha^−1^ of N fertilizer (B1N1) compared to other treatments. Field capacity (FC) and wilting point (WP) at −100 hPa to −15,500 hPa suction at 5–10 cm and 15–20 cm depths under biochar treatments were enhanced by 9–11% compared to non-biochar treatments (Figure 3A,B). However, soil FC and WP at 5–10 cm and 15–20 cm depths were significantly (*p* ≤ 0.05) higher (12–15%) after treatments B1N2 and B1N1 compared to the other treatments (Figure 3C,D). The factorial interactions (A*B and A*B*C) showed non-significant variations among all the treatments; only factors A and B significantly (*p* ≤ 0.05) enhanced VWC. This reflects the fact that biochar application has the potential to enhance VWC in soil.

The application of biochar increased the soil macropores at both the 5–10 cm and 15–20 cm depths. Three applications, namely 25 t·ha 25 t·ha^−1^ of biochar with 120 kg·ha^−1^ of N fertilizer (B1N2), 25 t·ha 25 t·ha^−1^ biochar alone (B1N0), and 25 t·ha 25 t·ha^−1^ of biochar with 160 kg·ha^−1^ of N fertilizer (B1N1), substantially (*p* = ≤0.05) enhanced macropores by 33.33%, 37.5%, and 41.17%, respectively, at the 5–10 cm depth. The same trend was recorded at the 15–20 cm soil depth, where the above treatments enhanced soil macropores by 40%, 45.45%, and 53.84%, respectively. However, no significant effect was recorded on mesopores or micropores (Figure 4A).

Biochar application had no substantial impact on soil pore distribution at either the 5–10 or 15–20 cm depths (Figure 4B). However, 120 kg·ha^−1^ of N fertilizer application significantly (*p* ≤ 0.05) enhanced soil macropores compared to all other treatments at both the 5–10 and 15–20 cm depths. Factors A, B, A*B, and A*B*C substantially (*p* ≤ 0.05) enhanced soil macropores by 30–40%, and the results recorded were non-significant for the rest factors. Thus, swine-digestate-derived biochar with and without N fertilizer application had a positive effect on soil macropores.

Biochar application significantly (*p* ≤ 0.05) reduced the HC of soil by 35–40% at the 5–10 cm soil depth compared to the non-biochar treatments. However, biochar did not affect HC at the 15–20 cm soil depth (Figure 5). The antilog of K of the soil varied during the entire season and ranged from 2.5 to 3.3% in May and from 2.6 to 4.7% in August. However, certain factorial interactions (B and A*B*C) recorded non-significant variations. Thus, among all the treatments, factors A and A*B showed substantial (*p* ≤ 0.05) effects on soil HC.

### 3.3. Soil Carbon Sources

The results suggest that the SCS utilization rate was significant and directly proportional to microbial growth (Figure 5). Carboxylic acid was the leading SCS utilized, and amines were the least-utilized carbon source. The overall utilization of all the SCSs was increased in biochar-treated soil compared to non-biochar treatments, e.g., B1N1 enhanced carbohydrates by 24.1%, B1N0 enhanced carboxylic acid by 32.8%, B1N0 enhanced amino acids by 23.2%, and B1N1 enhanced amines by 6.5% (Figure 6). Among the factorial interactions, factors A, B, and A*B showed significantly (*p* ≤ 0.05) enhanced SCS utilization.

### 3.4. Soil Microbiological Activity

According to all of the diversity indices (average well color development (AWCD), richness (R), and the McIntosh Index (U)) analyzed for the samples incubated in the Biolog EcoPlate for 96 h, higher biodiversity rates were recorded in biochar-treated soil (Figure 7). However, treatment with B0N1 also significantly (*p* ≤ 0.05) enhanced soil biodiversity. Soil biodiversity was characterized by high metabolic activity. Initially, at 24 and 48 h, treatment with B0N1 and B1N0 substantially (*p* ≤ 0.05) enhanced the AWCD rate by 50 and 59%, respectively, compared to the control treatment. Later on, at 72 and 96 h, treatment with B0N1 and B1N2 also significantly (*p* ≤ 0.05) enhanced AWCD by 55–60% compared to the control treatment (Figure 6). Similarly, the R index rate was significantly (*p* ≤ 0.05) enhanced from 24–96 h for the treatments B0N1, B1N0, B1N1, and B1N2; the rate increased by 20–35% compared to the control treatment. The *U* index recorded was significantly (*p* ≤ 0.05) lower from 24 to 96 h for treatment B1N2; it was 20–30% lower compared to the control treatment (Figure 6). Carbohydrates, amines, and miscellaneous (MS) followed the same trend as that of the R index and were significantly influenced under biochar application. Factorial interactions (A, B, and A*B) significantly (*p* ≤ 0.05) increased soil microbial activity, whereas factor A*B*C recorded non-significant results.

### 3.5. Correlation between Soil Physical Properties and Carbon Sources

Looking at the trait interrelations between N0 (Figure 8A), N1 (Figure 8B), and N2 (Figure 8C) under B0 conditions, BD was found to be significantly positively correlated, while TP was significantly negatively correlated to amino acids under B0N0 conditions and had no correlation recorded under B0N1 or B0N2. Similarly, BD was substantially positively correlated, and TP was significantly negatively correlated, to amines under B0N1. In contrast, BD was significantly negatively correlated, and TP was significantly positively correlated, to the antilog of K under B0N0 and B0N1 conditions. FC was substantially positively correlated to R and H under B0N0 and B0N2 conditions and significantly positively correlated to amino acids under B0N0 conditions and to amines under B0N1 conditions. PAW was significantly positively correlated to CH and amino acids under B0N0 conditions and significantly positively correlated to amines under B0N1 conditions and to U under B0N2 conditions. Macropores were found to be significantly negatively correlated to amino acids under B0N0 conditions and to amines under B0N1 conditions. Micropores were found to be significantly positively correlated to amino acids under B0N0 conditions and to amines under B0N1 conditions. AWCD was found to be significantly positively correlated to polymers and amino acids and significantly negatively correlated to the antilog of K under B0N1 conditions.

With the addition of biochar treatments, the trait interrelations between N0 (Figure 8D), N1 (Figure 8E), and N2 (Figure 8F) revealed that BD was significantly positively correlated, while TP was significantly negatively correlated, to R, H, amino acids, and MS under B1N0 conditions. FC was significantly positively correlated to R and H under B1N0 conditions. PAW was significantly positively correlated to MS under B1N1 conditions. Macropores were found to be significantly negatively correlated to R and H under B1N0 and B1N1 conditions, while under B1N0 conditions, they were significantly positively correlated to mesopores and significantly negatively correlated to CH, polymers, and amino acids. Mesopores were found to be significantly negatively correlated to micropores, AWCD, R, H, U, CH, and amines under B1N0 conditions. Macropores were found to be significantly positively correlated to CH, amino acids, and amines under B1N0 conditions. AWCD, R, and H were found to be significantly positively correlated to MS under B1N1 conditions and significantly negatively correlated to the antilog of K under B1N2 conditions. D was found to be significantly negatively correlated to CA and polymers and significantly positively correlated to MS under B1N2 conditions. U and CH were significantly negatively correlated to the antilog of K under B1N2 conditions. CH was significantly negatively correlated to HC under B1N1 conditions. Polymers, amino acids, amines, and MS were found to be significantly negatively correlated to the antilog of K under B1N2 conditions. Amino acids were significantly positively correlated to MS under B1N0 conditions.

### 3.6. Principal Component Analysis

The purpose of employing principal component analysis (PCA) was to compare biochar and characterize the associations between the hydro-physical properties, SCSs, and indices. According to PCA, the two axes of PCA explained 90.4% and 4.3% of the total variations, respectively. MS, CH, amino acids, U, R, micropores, AWCD, and H were relatively clustered together. Likewise, polymers, amines, and CA were relatively clustered together. TP and macropores clustered together (Figure 9).

## 4. Discussion

### 4.1. Biochar Effect on Soil Hydro-Physical Properties

Current results revealed a significant decrease in BD under 25 t·ha^−1^ of biochar with fertilizer application (Figure 1). There were several reasons for the soil BD reduction that are associated with biochar properties such as active large surface area, particle size, porosity, as well as soil properties [44]. Additionally, biochar has the ability to form soil pores in combination with soil particles, which results in a decrease in BD [45]. Šimanský et al. (2018) [46] reported that a 20 t·ha^−1^ biochar application significantly improved the soil structure compared to the control treatment, even though no significant improvement in the soil structure was recorded for a low-dose biochar application (10 t·ha^−1^). Biochar and other organic matter have the potential to improve the physical condition of the soil [47,48]. Figure 2 shows the significant improvement in TP. Biochar particles contain hydroxyl and carboxyl groups on their surfaces that enable soil organic particles and minerals to form a soil structure [49,50]. Biochar acts as a substrate for soil fauna that, when mixed with the soil particles in earthworms’ digestive tracts, produce coprolites that improve soil porosity and ultimately lead to lower BD [45,51].

The results indicated that the soil water content varied significantly and was substantially enhanced under biochar application with hPa suction at different depths (Figure 2). The reason for such variation could be rainfall and drought conditions [52]. In another study, it was reported that the soil water potential under biochar application tended to increase during the wheat-growing period [53]. They attributed this to biochar, which enhanced soil evaporation and tended to increase the soil temperature. Additionally, biochar significantly increased the soil’s water-holding capacity due to the fact that its large surface area tended to enhance the volumetric water content of soil [45]. In line with the results of our study, Claire L. Phillips reported that 9−36 Mg ha^−1^ of conifer-wood- and wheat-straw-derived biochar both significantly enhanced soil porosity, which tended to substantially enhance soil volumetric water content and soil field capacity [54].

In the current study, swine-digestate manure-derived biochar significantly increased soil macropores but had no effect on meso- or micro-porosity at both of the depths recorded (Figure 3). Biochar amendment had a direct effect on soil porosity due to the high porosity of biochar and its other physical properties [55,56]. Additionally, the increase in soil microporosity could be attributed to the higher rate of biochar amendment. However, an increase in a certain level of biochar restricts the rate, affecting the soil pore size distribution [57,58]. Due to various amounts of soil organic carbon (SOC) in aggregate fractions, macropores are richer in SOC content [59]. Biochar substantially enhanced soil porosity, but the mechanisms still remain unknown. It was reported in a study that biochar application indirectly enhanced the macropore fraction, but the soil contained <3% of biochar internally, which could not explain the increase in porosity [60].

Soil hydraulic conductivity allows soil to transmit water and influences every soil, depending upon soil type and the amount of mineral and organic content in the soil [61]. In this study, soil HC increased in biochar-treated soil by 35–40% compared to non-biochar treatments (Figure 5). The increase in soil HC was influenced by the particle sizes of biochar and the soil [62]; thus, it could be attributed to biochar amendment due to the fact that the particle sizes of biochar were larger than those of the soil at the experimental site [45]. Similarly, in another study, it was stated that soil HC might be influenced by improved soil structure and by biochar having greater particle sizes than the soil, and vice versa [63,64]. However, several factors were involved in measuring the value of the antilog of K, e.g., soil pores, aeration within soil pores, etc. [45], due to which some of the values of the antilog of K 15–20 cm depths were non-significant.

### 4.2. Biochar Effect on Soil Carbon Sources and Indices

Figure 6 and Figure 7 reveal the utilization trends of six major kinds of substrate guilds. The carbohydrate consumption capacity in the soil was recorded as being higher. Moreover, amino acids, carboxylic acids, polymers, amines, etc., were consumed much more extensively, and there was a significant SCS utilization trend recorded. These results imply that biochar with organic N fertilizer can increase the utilization of SCS, which tends to increase soil microbial diversity [65]. Higher diversity often increases the consumption of different substrates compared to deep soil, where microbial diversity is restricted [66,67]. Additionally, it is indicated that the soil depth gradient reduced both nutrient availability and the oxygen rate, which had a negative effect on soil bacteria and the regulation of their metabolic process [68,69]. The average well color development (AWCD) is a vital index of soil microbiota usage of carbon sources and reflects the physiological functions of soil microbial diversity [69,70]. Thus, it may be proposed that some of the selected C-source consumption may have a positive influence on microbial functional diversity and their metabolic activity in soil.

Under non-biochar treatments, the following correlations were found to be negative; TP and macropores to amino acids and amines under B0N0 and B0N1 conditions, respectively. BD and AWCD to HC under B0N0 and B0N1 conditions (Figure 8A–C). These negative correlations may be attributed to the lack of organic matter and carbon concentration [71,72]. In contrast, under biochar-treated soil, the following correlations were recorded as positive: macropores, FC, and BD to R, H, CH, amino acids, amines, and MS under B1N0 conditions. Amino acids were significantly and positively correlated to MS under B1N0 conditions (Figure 7D–F). Several studies reported that biochar application exerted positive priming effects by stimulating the soil’s organic carbon content, which built a strong correlation with soil physicochemical properties [73,74].

## 5. Conclusions

The conclusions of this study are as follows:Biochar alone and applied with 160 kg·ha^−1^ and 120 kg·ha^−1^ of N fertilizer significantly reduced soil BD and enhanced TP, as well as substantially enhanced soil macropores at both studied depths during August. Thus, swine-digestate manure-derived biochar may be a useful amendment to soil facing the problem of high BD and low TP, as well as in compacted soil with lower soil porosity.Biochar with 160 kg·ha^−1^ of N fertilizer substantially increased VWC at the 5–10 cm depth at −4 to −100 hPa suction, whereas at higher suction (−100 hPa to −15,500 hPa), both field capacity and the wilting point of soil were recorded as being higher at both the 5–10 and 15–20 cm depths. Thus, biochar application may be helpful in drought conditions to enhance soil water content.Biochar with and without N fertilizer application significantly lowered soil hydraulic conductivity by 35–40% at the 5–10 cm depth compared only to the non-biochar treatments. Thus, swine-digestate manure-derived biochar may substantially improve water transmission within the topsoil layer.Biochar amendment may substantially enhance carbon source utilization, which tends to enhance soil microbial activity and was positively correlated in this study. Carboxylic acid was the leading SCS utilized, and amines were the least-utilized carbon source. The overall utilization of all SCSs was increased in biochar-treated soil compared to non-biochar treatments. According to all of the diversity indices (e.g., average well color development (AWCD) and richness (S)) analyzed in the Biolog EcoPlate incubated for 96 h, with the exception of the Mclntosh Index (U), higher biodiversity rates were recorded in biochar-treated soil and with the B0N1 treatment. However, the U index was recorded as being significantly lower from 24 to 96 h under treatment with B1N2; it was 20–30% lower compared to the control treatment. This study summarized that swine-digestate manure-derived biochar, both with and without N fertilizer, may be a useful amendment; depending upon the type of soil and the environmental factors, it may be useful in improving hydro-physical properties and microbial abundance.

## Figures and Tables

**Figure 1 plants-11-01729-f001:**
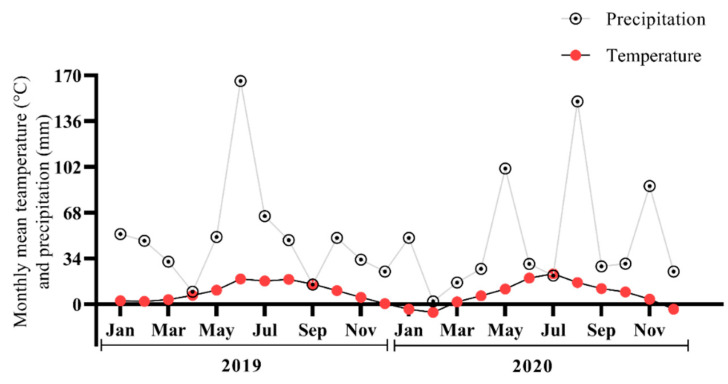
Monthly mean temperature and precipitation at the experimental site during the period 2019–2020.

**Figure 2 plants-11-01729-f002:**
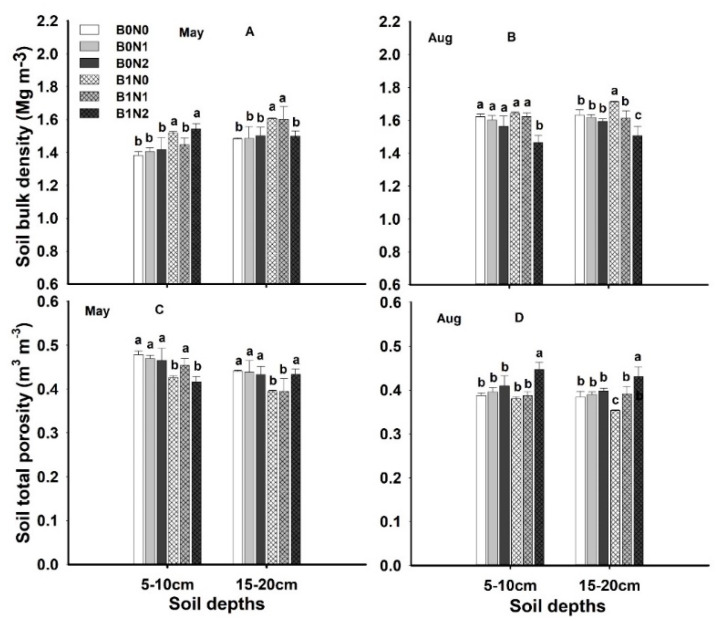
Effect of different treatments onsoil bulk density (**B**,**D**), (**A**) in May at 5–10 cm and 15–20 cm depth, (**B**) in August at 5–10 cm and 15–20 cm depth, and total soil porosity (**C**) in May at 5–10 cm and 15–20 cm depth, (**D**) in August at 5–10 cm and 15–20 cm depth. The letters a, b, c indicate statistically significant difference at *p* < 0.05.

**Figure 3 plants-11-01729-f003:**
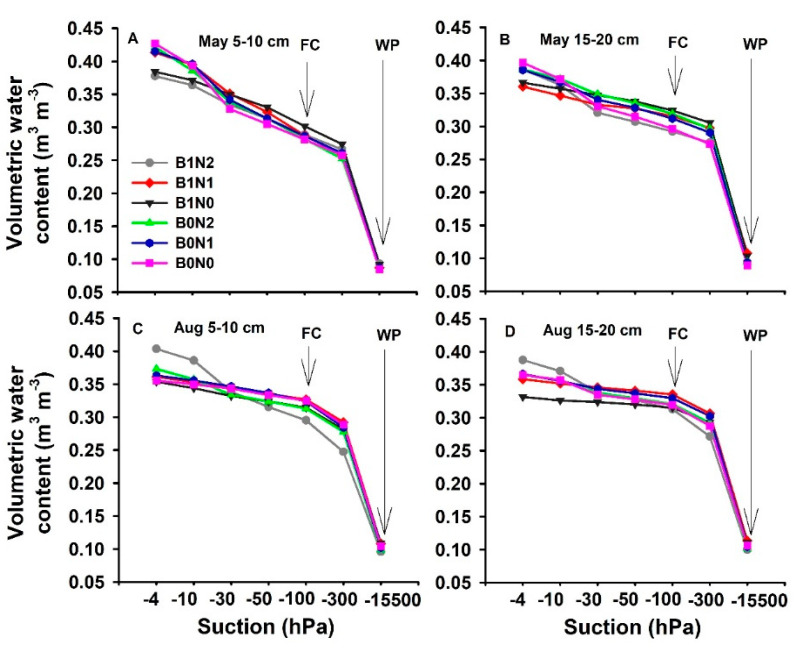
Effect of different treatments on VWC (volumetric water content), FC (field capacity), and WP (plant wilting point) (**A**) in May at 5–10 cm depth, (**B**) in May at 15–20 cm depth, (**C**) in August at 5–10 cm depth, (**D**) in August at 15–20 cm depth.

**Figure 4 plants-11-01729-f004:**
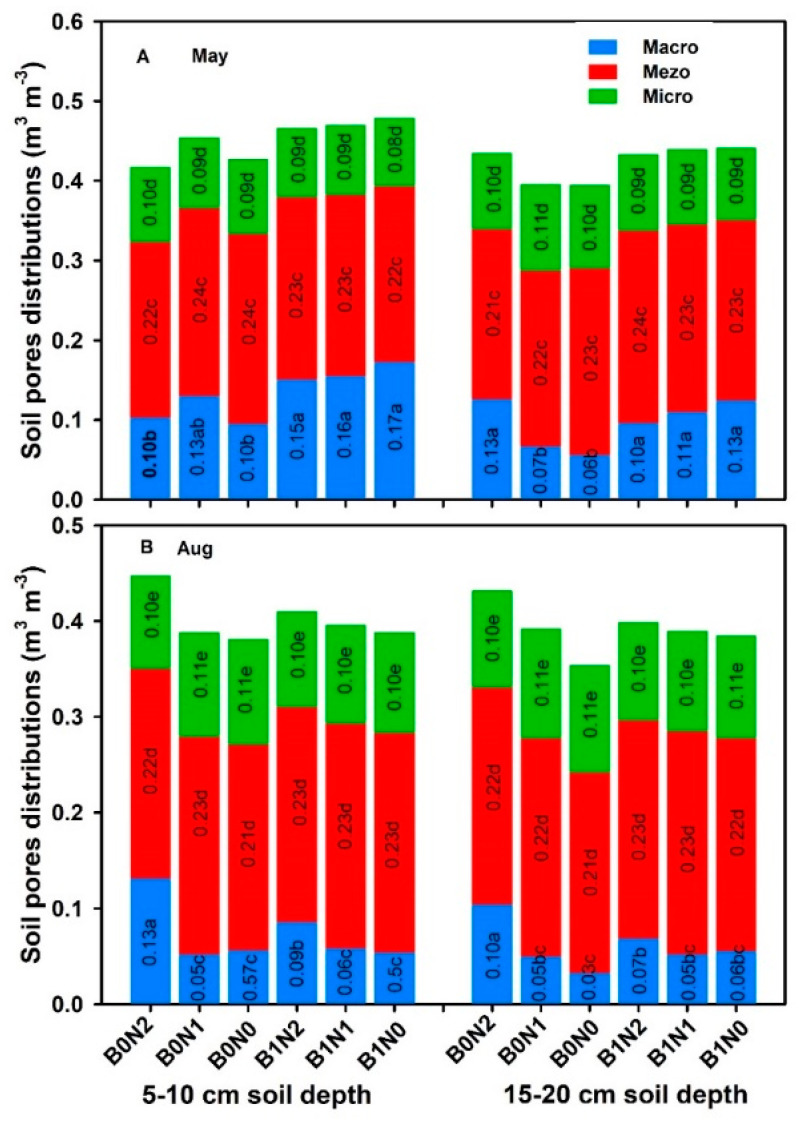
Effect of different treatments on pore size distribution (macropores, mesopores, and micropores) (**A**) in May, at 5–10 cm and 15–20 cm depths, (**B**) in August at 5–10 cm and 15–20 cm depths for B0N0 (without biochar or N fertilization); B0N1 (without biochar and with 160 kg·ha^−1^ N); B0N2 (without biochar and with 120 kg·ha^−1^ N); B1N0 (biochar 25 t·ha 25 t·ha^−1^ only); B1N1 (biochar 25 t·ha^−1^ and 160 kg·ha^−1^ N); and B1N2 (biochar 25 t·ha^−1^ and 120 kg·ha^−1^ N. The letters a–e indicate statistically significant difference at *p* < 0.05.

**Figure 5 plants-11-01729-f005:**
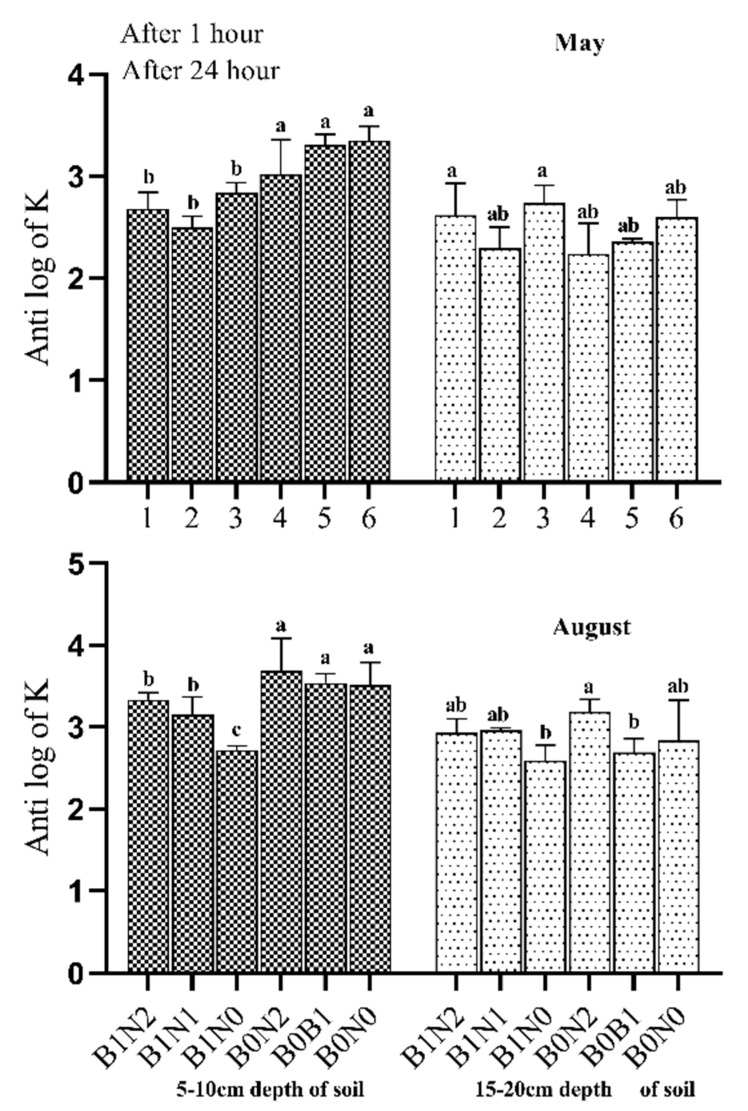
Effect of different treatments on soil hydraulic conductivity. B0N0 (without biochar or N fertilization); B0N1 (without biochar and with 160 kg·ha^−1^ N); B0N2 (without biochar and with 120 kg·ha^−1^ N); B1N0 (biochar 25 t·ha^−1^ only); B1N1 (biochar 25 t·ha^−1^ and 160 kg·ha^−1^ N); and B1N2 (biochar 25 t·ha^−1^ and 120 kg·ha^−1^ N) at two different depths (5–10 cm and 15–20 cm) and times (May and August). The letters a, b, c indicate statistically significant difference at *p* < 0.05.

**Figure 6 plants-11-01729-f006:**
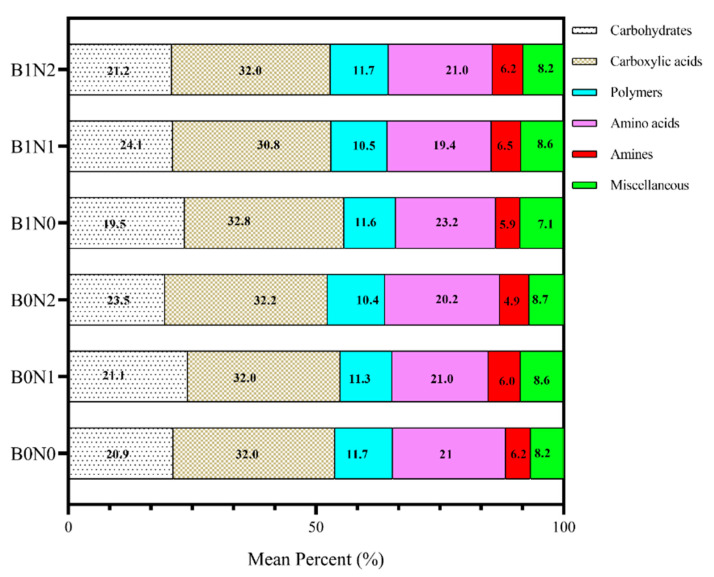
Effect of different treatments on the average mean of soil carbon sources. B0N0 (without biochar or N fertilization); B0N1 (without biochar and with 160 kg·ha^−1^ N); B0N2 (without biochar and with 120 kg·ha^−1^ N); B1N0 (biochar 25 t·ha^−1^ only); B1N1 (biochar 25 t·ha^−1^ and 160 kg·ha^−1^ N); and B1N2 (biochar 25 t·ha^−1^ and 120 kg·ha^−1^ N).

**Figure 7 plants-11-01729-f007:**
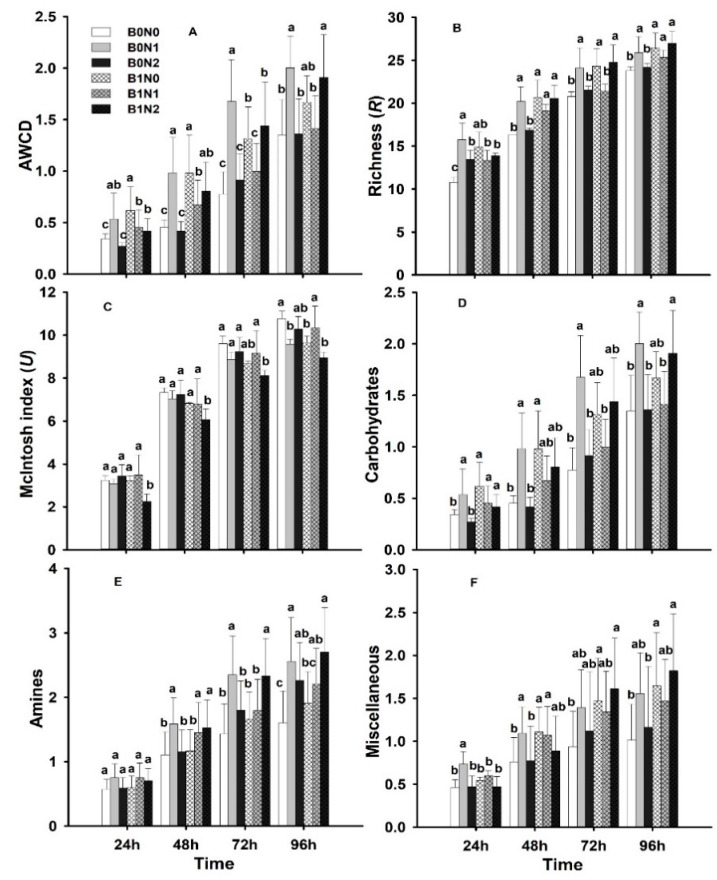
Effect of different treatments on soil carbon sources (**A**) Average well color development (AWCD), (**B**) Richness (R), (**C**) Mclntosh index (*U*), (**D**) Carbohydrates, (**E**) Amines, (**F**) Miscellaneous; B0N0 (without biochar or N fertilization); B0N1 (without biochar and with 160 kg·ha^−1^ N); B0N2 (without biochar and with 120 kg·ha^−1^ N); B1N0 (biochar 25 t·ha^−1^ only); B1N1 (biochar 25 t·ha^−1^ and 160 kg·ha^−1^ N); B1N2 (biochar 25 t·ha^−1^ and 120 kg·ha^−1^ N) at different times (24, 48, 72, and 96 h). The letters a, b, c, indicate statistically significant difference at *p* < 0.05.

**Figure 8 plants-11-01729-f008:**
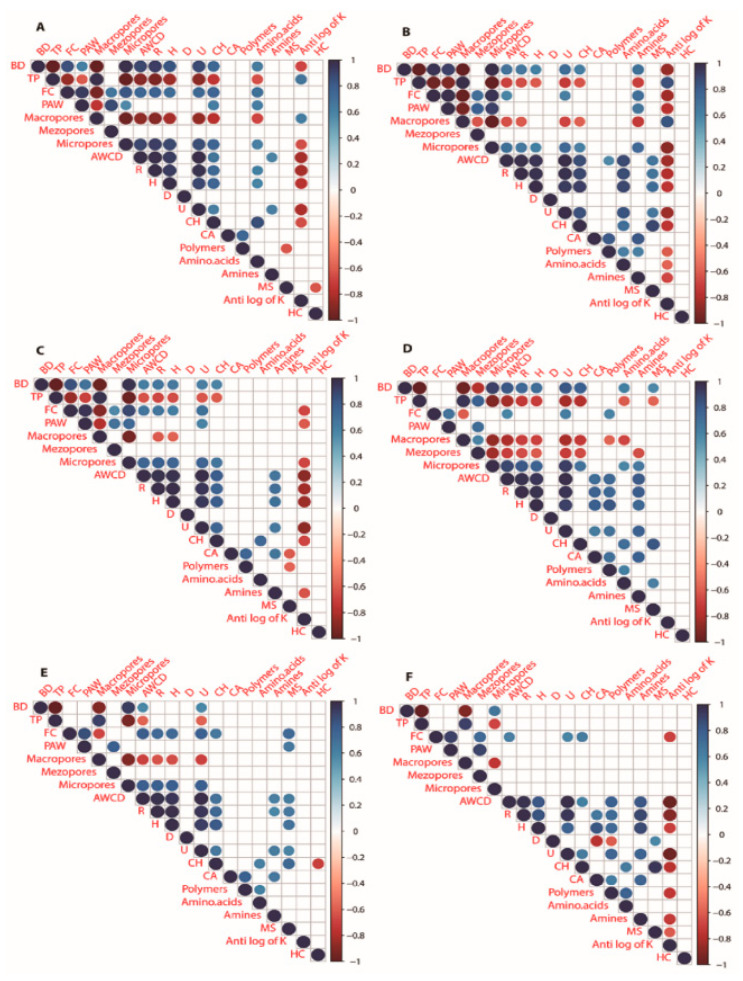
Heatmap correlations under different treatments (**A**) B0N0 (without biochar and N fertilization); (**B**) B0N1 (Without Biochar and 160 kg·ha^−1^ N); (**C**) B0N2 (Without Biochar and 120 kg·ha^−1^ N fertilization); (**D**) B1N0 (Biochar 25 t·ha^−1^ only); (**E**) B1N1 (Biochar 25 t·ha^−1^ and 120 kg·ha^−1^ N); (**F**) B1N2 (Biochar 25 t·ha^−1^, 160 kg·ha^−1^ N) for bulk density (BD), total porosity (TP), field capacity (FC), plant-available water (PAW), average well color development (AWCD), richness (R), the Shannon index (H), the Simpson index (D), the Mclntosh index (U), carbohydrates (CH), carboxylic acid (CA), miscellaneous (MS), hydraulic conductivity (HC). Note: Significant (*p* < 0.05) negative (red color) and positive (blue color) correlations between different soil carbon sources and soil physical properties are identified by color (−1.0 to +1.0); non-significant correlations are omitted.

**Figure 9 plants-11-01729-f009:**
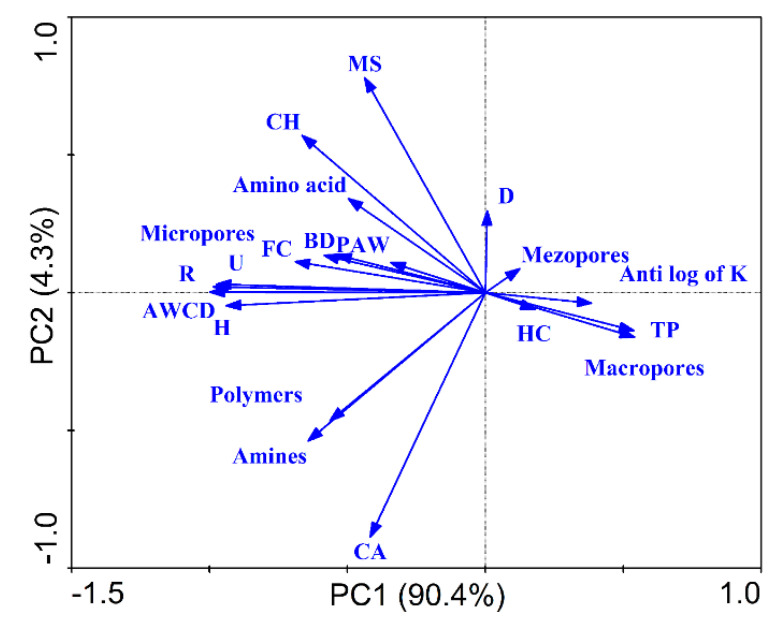
Pearson’s correlations for bulk density (BD), total porosity (TP), field capacity (FC), plant-available water (PAW), average well color development (AWCD), richness (R), the Shannon index (H), the Simpson index (D), the Mclntosh index (U), carbohydrates (CH), carboxylic acid (CA), miscellaneous (MS), and hydraulic conductivity (HC) under different treatments, i.e., B0N0 (without biochar or N fertilization); B0N1 (without biochar and with 160 kg·ha^−1^ N); B0N2 (without biochar and with 120 kg·ha^−1^ N); B1N0 (biochar 25 t·ha^−1^ only); B1N1 (biochar 25 t·ha^−1^ and 160 kg·ha^−1^ N); and B1N2 (biochar 25 t·ha^−1^ and 120 kg·ha^−1^ N).

**Table 1 plants-11-01729-t001:** Physicochemical characteristics of the soil and biochar under experimental trial.

Physicochemical Properties	Soil	Biochar
pH_KCl_	7.5	9.1
Ash content (%)	-	32.21
Moisture wt. (%)	-	2.52
Volatiles wt. (%)	-	56.73
Residual mass (char formed) wt. (%)	-	40.75
Total N (g/kg)	0.01	19.18
Ammonium N (mg/kg)	1.21	-
Mineral N (mg/kg)	11.21	-
Available P (g/kg)	0.145	-
Available K (g/kg)	0.213	-
Total Mg (g/kg)	-	10.50
Organic C (%)	1.10	62.33

## Data Availability

All datasets generated for this study are included in the article.

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
