# Peer review of "The Impact of Swine Manure Biochar on the Physical Properties and Microbial Activity of Loamy Soils"

_plants, 2022, doi:10.3390/plants11131729_

Round 1

Reviewer 1 Report

The abstract and the keywords are well chosen and describe the correct entire manuscript. The introduction is incorrectly edited. Citations are numbered once and once as the author's name and the year's name - please correct it. Please consider the importance of the literature. In the methodology: I do not understand line 109-111. When using analytical devices, please provide their rpducent and the standard according to which the measurements are made, eg TGA. Line 112 is related to torrefaction? please explain. Please separate the soil and biochar sections in the methodology. I am asking for detection limits, standards and recoveries for the methods of testing elements - especially the assimilable forms. Was extraction used? how? Yes, what? Were the samples mineralized and in what acid? The results are legible and clear except for Fig 4 - not legible to me - fuzzy black letters on the red bar of the result. Fig 5 - to sharpen - not legible. Fig 6 - unreadable results on the result bar. Fig 8 - Descriptions are not legible. Discussion and Conclusions - View and Acceptable. Please check English with a native speaker.

Author Response

Thank you so much for the intellectual review. This was helpful to improve my manuscript.

I have considered all the comments and suggestions and tried my best to respond to all comments as explained below and changes highlighted in the manuscript.

I have changed the whole introduction section as there were several citation mistakes. Please see the section.

Sorry for the inconvenience of not getting lines 109-111, which are all about TGA. I have explained the TGA process that how it works and how I used it.  It’s a bit more explanation of the process could be helpful for the reader (my thoughts). I have also added the reference that shows TGA  previously being used.

Line 112 is also the part of TGA process. Line (109-112) now 115-117 shows the whole mechanism of TGA. It describes how TGA works. I am not very much sure about the term  “torrefaction” or whether the pyrolysis being performed during the TGA process is called torrefaction or not.

In the methodology section, all the soil properties are separately analyzed like Hydraulic conductivity, carbon sources, indices pore size distribution, etc. Just a physicochemical property of a couple of samples of biochar was analyzed in the same way as soil samples were done, and it was also done separately. ICP was used for DTPA extractable nutrients P, K, Ca, and Mg, in which the soil sample was introduced as a liquid using a nebulizer and spray chamber. The nebulizer uses the supersonic expansion of gas to turn the liquid into a fine mist, and the spray chamber then removes any droplets that are too large to be processed in the plasma. Similarly pH and EC of the soil and Biochar.

All the figures are resized and made clear.

Reviewer 2 Report

19: "B1 = 25t ha-1" - The correct notation is B1 = 25 Mg‧ha-1 . Please apply throughout the work.

34: "In recent years, biochar has been used extensively as a soil conditioner to improve soil quality (1)." I suggest quoting the literature as [1].

72: "(21) The." Please put a period.

100: Introduction chapter should end in aim.

106: "550 â—¦C" Remove the space.

112: "60 ml / min." The correct notation is 60 ml‧min-1.

134: "vol vol-1" ???

152: "(2 ° C).". Remove the space.

184: Please insert a pattern, do not paste

206” „Table 1. Physiochemical characterises of soil and Biochar under experimental trial.”. Why is missing so much data? Please complete them. This is extremely important information.

206” „Table 1. Physiochemical characterises of soil and Biochar under experimental trial.”. Please add information about the digestate.

226: „Figure 1.”. Please standardize the axis descriptions.

Author Response

Thank you so much for the scholarly review.

I have considered all the comments and suggestions and tried to respond below and highlighted them in the manuscript.

Reviewer’s comment: 19: "B1 = 25t ha-1" - The correct notation is B1 = 25 Mg‧ha-1 ”. Please apply throughout the work.

Author’s Response: Done

Reviewer’s comment: 34: "In recent years, biochar has been used extensively as a soil conditioner to improve soil quality (1)." I suggest quoting the literature as [1].

Author’s Response: Done

Reviewer’s comment: 72: "(21) The." Please put a period.

Author’s Response: Done

Reviewer’s comment: 100: The introduction chapter should end in aim.

Author’s Response: Done

Reviewer’s comment: 106: "550 â—¦C" Remove the space.

Author’s Response: Done

Reviewer’s comment: 112: "60 ml / min." The correct notation is 60 ml‧min-1.

Author’s Response: Done

Reviewer’s comment: 134: "vol vol-1" ???

Author’s Response: Done

Reviewer’s comment: 152: "(2 ° C).". Remove the space.

Author’s Response: All done

Reviewer’s comment: 184: Please insert a pattern, do not paste

Author’s Response: All done

Reviewer’s comment: 206” „Table 1. Physiochemical characterizes of soil and Biochar under experimental trial.”. Why is missing so much data? Please complete them. This is extremely important information.

Author’s Response: The missing data in the table does not exist for soil, for instance, the soil doesn’t have char content, ash content, and volatiles as compared to Biochar.

Reviewer’s comment: 206” „Table 1. Physiochemical characterizes of soil and Biochar under experimental trial.”. Please add information about the digestate.

Author’s Response: Actually, we didn’t analyze the digestate properties, as we converted the digestate into biochar and have no direct application of digestate into the field but rather deal with biochar.

Reviewer’s comment: 226: „Figure 1.”. Please standardize the axis descriptions.

Author’s Response: Done

Reviewer 3 Report

- change title to : impact of Swine manure biochar on soil physical properties and microbial activity of loamy soils

- line 42-43 type of soils

- line 94 define SCS

- loamy soils

-209-211 delete as this part was repeated in other section

Author Response

We are very much thankful for your precious comments and suggestions.

We have considered all the comments and tried to respond to all below and highlighted in the manuscript.

Reviewer comments: change title to : impact of Swine manure biochar on soil physical properties and microbial activity of loamy soils

Author's response: Done

Reviewer comments: line 42-43 type of soils

Author's response: Done

Reviewer comments:-209-211 delete as this part was repeated in other section

Author's response: Actually this is to show what is factor A, B and C. In case, if we delete it, the reader may not understand the 3-way ANOVA/Statistic applied. If you still recommend it, I will delete it in proofreading.